# Seroprevalence and related risk factors of Brucella spp. in livestock and humans in Garbatula subcounty, Isiolo county, Kenya

Athman Mwatondo[1,2,3]*, Mathew Muturi[2,4,5], James Akoko[2], Richard Nyamota[2], Daniel Nthiwa[6], Josphat Maina[3], Jack Omolo[7], Stephen Gichuhi[8], Marianne W. Mureithi[1], Bernard Bett[2]

1 Department of Medical Microbiology and Immunology, University of Nairobi, Nairobi, Kenya, 2 International Livestock Research Institute, Nairobi, Kenya, 3 Zoonotic Disease Unit, Ministry of Health, Nairobi, Kenya, 4 Dahlem Research School of Biomedical Sciences, Department of Veterinary Medicine, Freie Universität Berlin, Berlin, Germany, 5 Zoonotic Disease Unit, Ministry of Agriculture, Livestock and Fisheries, Nairobi, Kenya, 6 Department of Biological Sciences, University of Embu, Embu, Kenya, 7 County Government of Kilifi, Department of Agriculture, Livestock Development and Fisheries, Kilifi, Kenya, 8 Department of Ophthalmology, University of Nairobi, Nairobi, Kenya

* amwatondo@yahoo.com

**Data Availability Statement:** All relevant data are within the manuscript and its Supporting Information files.

## Abstract

### Background

Brucellosis is a neglected zoonotic disease that affects both animals and humans, causing debilitating illness in humans and socio-economic losses in livestock-keeping households globally. The disease is endemic in many developing countries, including Kenya, but measures to prevent and control the disease are often inadequate among high-risk populations. This study aimed to investigate the human and livestock seroprevalence of brucellosis and associated risk factors of Brucella spp. in a pastoralist region of northern Kenya.

### Methods

A cross-sectional survey was conducted using a two-stage cluster sampling method to select households, livestock, and humans for sampling. Blood samples were collected from 683 humans and 2157 animals, and Brucella immunoglobulin G (IgG) antibodies were detected using enzyme-linked immunosorbent assays. A structured questionnaire was used to collect data on potential risk factors associated with human and animal exposures. Risk factors associated with Brucella spp. exposures in humans and livestock were identified using Multivariate logistic regression.

### Results

The results indicated an overall livestock Brucella spp. seroprevalence of 10.4% (95% Confidence Interval (CI): 9.2–11.7). Camels had the highest exposure rates at 19.6% (95% CI: 12.4–27.3), followed by goats at 13.2% (95% CI: 9.3–17.1), cattle at 13.1% (95% CI: 11.1–15.3) and sheep at 5.4% (95% CI: 4.0–6.9). The herd-level seroprevalence was 51.7% (95% CI: 47.9–55.7). Adult animals (Adjusted Odds Ratio (aOR) = 2.3, CI: 1.3–4.0), female

**Funding:** This study was implemented under the project 'Co-infection with Rift Valley fever virus, Brucella spp. and Coxiella burnetii in humans and animals in Kenya: Disease burden and ecological factors' funded by Defense Threat Reduction Agency, Grant No. HDTRA11910031 to BB. We also acknowledge the CGIAR Fund Donors (https://www.cgiar.org/funders). The funders had no role in study design, data collection and analysis, decision to publish, or preparation of the manuscript. The funders website is https://www.dtra.mil/.

**Competing interests:** The authors have declared that no competing interests exist.

animals (aOR = 1.7, CI: 1.1–2.6), and large herd sizes (aOR = 2.3, CI: 1.3–4.0) were significantly associated with anti-brucella antibody detection while sheep had significantly lower odds of *Brucella* spp. exposure compared to cattle (aOR = 1.3, CI: 0.8–2.1) and camels (aOR = 2.4, CI: 1.2–4.8). Human individual and household seroprevalences were 54.0% (95% CI: 50.2–58.0) and 86.4% (95% CI: 84.0–89.0), respectively. Significant risk factors associated with human seropositivity included being male (aOR = 2.1, CI:1.3–3.2), residing in Sericho ward (aOR = 1.6, CI:1.1–2.5) and having no formal education (aOR = 3.0, CI:1.5–5.9). There was a strong correlation between human seropositivity and herd exposure (aOR = 1.6, CI:1.2–2.3).

## Conclusions

The study provides evidence of high human and livestock exposures to *Brucella* spp. and identifies important risk factors associated with disease spread. These findings emphasize the need for targeted prevention and control measures to curb the spread of brucellosis and implement a One Health surveillance to ensure early detection of the disease in Isiolo County, Northern Kenya.

## Author summary

Brucellosis is a priority zoonotic disease in Kenya that causes human illness and socio-economic losses in livestock-keeping households. We conducted a linked human-livestock study to investigate the seroprevalence of *Brucella* spp.. We also identified risk factors associated with brucellosis seropositivity. The study found a high seroprevalence in humans and animals and identified significant risk factors associated with livestock exposure, which included adult animals, female animals, and large herd sizes. In humans, significant risk factors included being male, residing in a specific ward, and having no formal education. We also found a strong correlation between human seropositivity and herd exposure. Implementing interventions such as promoting public awareness about the disease, safe handling of livestock and livestock products, and livestock vaccinations can potentially reduce the disease transmission in livestock and spill over to humans. Future studies should examine the efficacy of measures that reduce these risk factors to mitigate the spread of brucellosis.

## Introduction

Brucellosis is a neglected zoonotic disease that causes debilitating illnesses in humans and extensive socio-economic losses in livestock-keeping households globally [1, 2]. Six classical *Brucella* species infect animals, but only four have zoonotic potential; *B. abortus*, whose primary host is cattle; *B. melitensis*, which infects goats and sheep and *B. suis* and *B. canis*, which infects pigs and dogs respectively [2–4]. Domesticated animals, such as cattle, sheep, goats, camels, and pigs, are the primary sources of human infections, while wild animals may act as reservoirs in areas with livestock-wildlife interactions [5]. Human infections are primarily attributed to *B. melitensis*, the most virulent species [6]. Human exposure is principally through consumption of unpasteurized milk and dairy products, undercooked meat and meat products and direct contact with an infected aborted fetus, placenta, fetal fluids, and vaginal

discharges [7]. High-risk groups include livestock keepers, field animal health workers, abattoir workers, dairy industry workers, and laboratory professionals [8, 9]. Human-to-human transmission of brucellosis is rare, but cases have been reported through transplacental infection, breastfeeding, and organ transplantation [10].

Brucellosis is a systemic disease in humans, mainly presenting as a febrile illness with acute, subacute, or chronic forms depending on the stage of the disease [11, 12]. The disease is characterized by recurrent, undulant fevers, arthralgia, myalgia, malaise, sweating, lower back pain and headache [8, 13]. The symptoms are often accompanied by localized single or multiple-organ involvement [12]. Due to the unspecific nature of the signs and symptoms of brucellosis, diagnosis requires assessment of a combination of clinical, epidemiological and laboratory criteria. Multiple diagnostic methods exist for detecting *Brucella* bacterial infections. These methods include serological tests such as Rose Bengal Test (RBT), Standard Tube Agglutination Test (SAT), and Enzyme-Linked Immunosorbent Assay (ELISA) that identify *Brucella*-specific antibodies in the blood. Confirmatory techniques include bacterial culture and Polymerase Chain Reaction (PCR), which require specialized equipment and expertise, making them less feasible in most endemic settings [14]. Treating brucellosis is costly, as it often requires a prolonged course of antibiotic therapy. Furthermore, treatment may be complicated by factors such as treatment failure, relapses, and severe side effects from the drug regimen [12]. The disease is rarely fatal, with a less than two percent fatality rate in untreated cases. However, if left untreated or inadequately treated, it can lead to long-term consequences such as chronic infections, infertility, arthritis, endocarditis, and neurological complications [15].

Transmission in livestock is mainly through ingestion of pastures or water contaminated by-products of abortion or parturition such as placental membranes and vaginal fluids [16, 17]. Other modes of transmission include ingesting contaminated feed, water, or vegetation that has been contaminated with *Brucella* bacteria. Additionally, sexual transmission can occur from infected males to females during mating or artificial insemination, and there is also vertical transmission, where the disease is passed from an infected mother to her offspring during pregnancy. This can lead to abortion, stillbirths, or the birth of weak offspring. Infection in most livestock species presents as a reproductive disorder characterized by infertility, abortion, reduced newborn survival rates, and decreased milk production. As such, the disease is associated with significant direct economic losses to farmers and indirect losses in endemic countries due to restricted trade in livestock and livestock products [18–20].

Brucellosis has been controlled in most developed countries, but it is still prevalent in Asia, Africa, Central and South America, and the Middle East [20], with approximately 500 million human infections reported annually. The incidence is much higher, with estimates ranging from 10 to 25 times the reported numbers [21]. In Kenya, brucellosis is endemic and ranked the fourth most important zoonotic disease [22]. However, its actual burden on livestock and human populations remains inadequately described due to weak surveillance, low utilization of health care services, and the widespread use of non-prescribed antibiotics among affected communities [23–25]. Studies have demonstrated that the disease has been prevalent in pastoralist communities in the last ten years, with seroprevalence estimates ranging from 10% to 47% in apparently healthy humans and 3.4% to 22.3% in livestock [23–27].

The prevention and control of brucellosis in endemic low- and middle-income countries (LMICs) require integrated One Health approaches, including integrated activities in research, surveillance, and response [21, 28, 29]. This study was conducted at the interface between humans and livestock in Isiolo County as part of a larger investigation into co-exposures of Rift Valley fever virus, *Brucella* spp., and *Coxiella burnetii* in both humans and animals in northern Kenya. Our objectives were two-fold; to estimate the seroprevalence of *Brucella* spp. and to determine the risk factors associated with *Brucella* spp. seropositivity. The results of this

study will provide valuable evidence and baseline data for establishing an integrated livestock-human surveillance system in the study area.

## Materials and methods

### Ethics statement

We obtained ethical approval for human and livestock sampling from the Institutional Research Ethics Committee at the International Livestock Research Institute (reference number: ILRI-IREC2020-07). The household heads provided written consent for livestock sampling, and written consent was obtained from all selected humans for sampling. The study clinicians informed all participants that their participation was voluntary, and they could decline without any consequences. For adults (aged >18 years), written consents were provided; for children aged between 13–17 years, written assent forms and parental/guardian permissions were obtained. For children under 13 years, only written parental/guardian permissions were obtained.

### Study area

The study was conducted in Garbatula subcounty in Isiolo County. Isiolo County is in the northern region of Kenya (Fig 1). The subcounty is subdivided into three administrative wards: Garbatulla, Kinna and Sericho. The study area is in a relatively low-lying region, with Kinna ward having the highest altitude of 589 meters (m) and the lowest area being Sericho at 459m. The area receives an average of 580 millimetres of rainfall per annum, with November and April being the wettest months. The area is characterized by hot and dry climate for most of the year, with a mean annual temperature of 29˚C, with variations due to differences in altitude [30].

Garbatula subcounty is sparsely populated, with an average population density of nine people per square kilometre [31]. The Borana tribe is the most predominant, and livestock is a source of livelihood for over 80% of the inhabitants [32]. Cattle, sheep, goats, and camels are the main livestock species raised in the area, although goats and sheep have the highest population. The subcounty shares a border with Meru National Park and the Bisanadi Game Reserve, two unconfined sanctuaries for wild animals. As a result, there are frequent interactions between domestic livestock and wild animals in certain regions of the subcounty.

### Sample size determination

The sample size (n) for humans and animals (camel, cattle, sheep, and goats combined) was estimated using the formula; $n = \frac{1.96^2 * p(1-p)}{d^2}$ [33]. The expected (a priori) seroprevalence estimate (p) of Brucella species was assumed to be 50% in both humans and animals since there were no studies on Brucella burden from the area. It was further assumed that the study would estimate this outcome with an error margin (d) of 0.05.

Previous studies have shown that Brucella spp. seroprevalence clusters in humans and animals at the household and herd levels. A design effect (Deff) was therefore determined for adjusting the naïve sample sizes by assuming that an average of twenty animals per herd and three humans per household would be sampled and that Brucella spp. seroprevalences have intracluster correlation coefficients (ICC) of 0.2 in animals and 0.21 in humans, respectively [24]. DEFF estimates of 4.8 and 1.42 were derived for animals and humans, respectively. Minimum sample sizes of 546 humans and 1844 animals were therefore estimated based on multiplying the DEFF estimates with the naïve sample sizes.

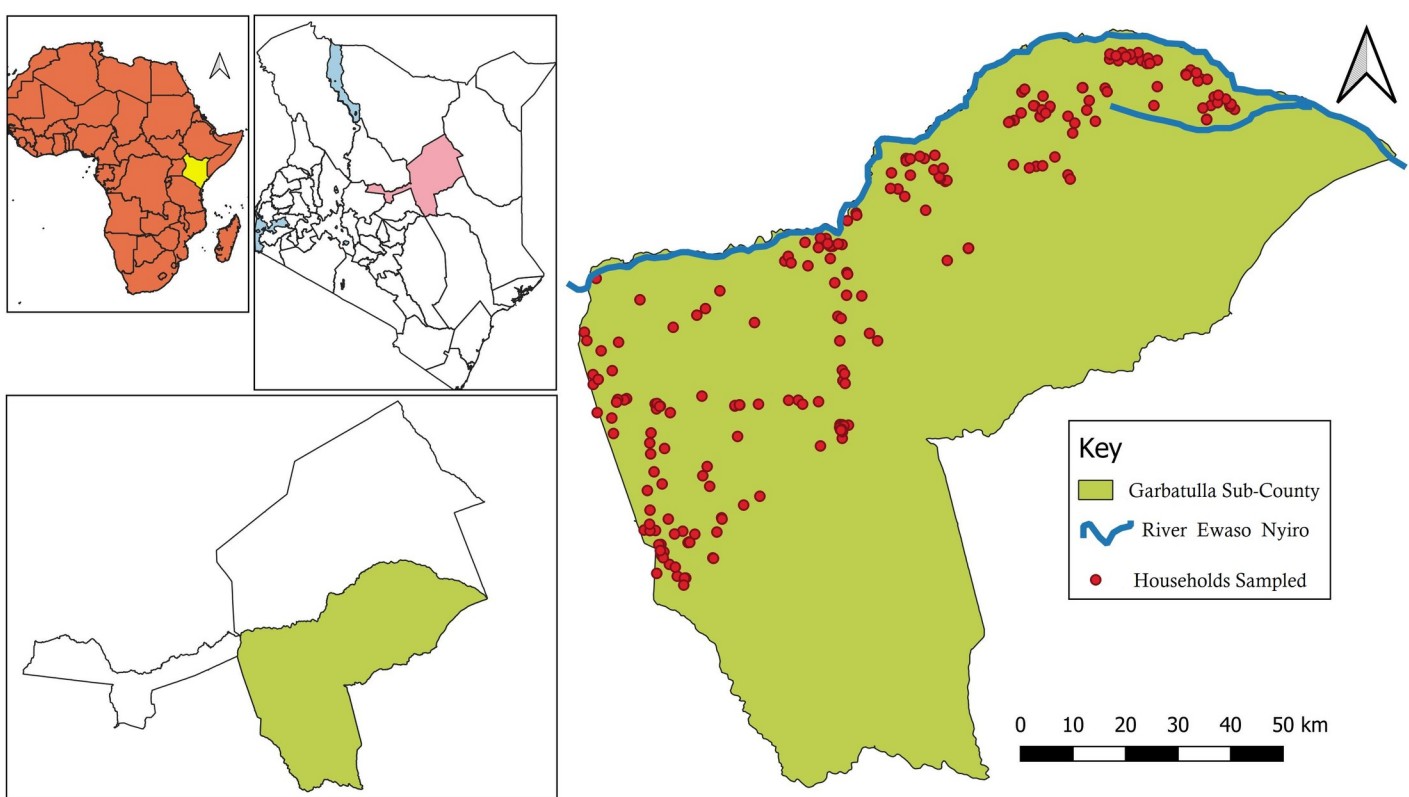

**Fig 1. Map of Africa and Kenya showing sampling sites within Garbatula subcounty (*The underlying map layer used to generate this figure was obtained from https://www.diva-gis.org/gdata*).**

## Sampling design for livestock and humans

Livestock and human data were collected in July-August 2021 in a cross-sectional, two-stage random sampling design. The primary sampling units were individual livestock and humans within randomly selected households. Households were eligible for selection if they kept or owned any of the four livestock species and had at least one consenting adult human. To identify study households, 250 random geographical coordinates (RGC) were generated using Quantum Global Information System (QGIS) version 3.4.4 software (https://qgis.org), for Garbatula Subcounty and households closest to an RGC were selected for sampling. For the coordinates that fell near a cultural homestead (Manyatta) where livestock from different households are kept in one enclosure, one household was randomly selected, and only humans and animals owned by the household were eligible for sampling.

A household was defined as a group of people using a common cooking area. A total of five animals were selected per species (camel, sheep, cattle, and goats) in each herd using systematic random sampling. Animals under one month and the very sick were excluded from the study. In herds with fewer than five animals per species, all animals were selected. All household members were eligible for enrolment, excluding children under two years since the ethical approval did not cover them.

## Sample collection in livestock and humans

Qualified veterinarians collected livestock samples. During sampling, animals were restrained by trained animal handlers. Using aseptic techniques, blood was collected from the jugular

vein of each animal into 10 ml plain vacutainers with unique barcode labels. The vacutainers were stored in cool boxes with icepacks to maintain temperatures of 4–8˚C in the field before transfer to the local field veterinary laboratory for serum extraction and temporary storage at -20˚C. The samples were centrifuged at 2500 gravitational force for 15 minutes, and the serum was stored in 2 ml barcoded cryovials. Extracted serum samples were transported at -20˚C using motorized freezers to the International Livestock Research Institute (ILRI) Laboratories in Kenya every five days, where they were first stored at -20˚C before testing for antibodies to *Brucella* species and then transferred to -80˚C for long-term storage. Qualified clinicians conducted the sampling in humans. Each participant had 5 mL of blood drawn from their median cubital vein on the left hand using plain vacutainers with unique barcode labels. All blood samples were immediately placed in a cool box and maintained at 4–8˚C in the field before being transferred to nearby health centres, within four hours. The procedures for extracting serum, storing and transporting samples were identical to those used for animal samples. Testing of human samples was conducted at ILRI.

## Data collection

Data was collected by the trained study staff using standardized questionnaires installed in Open Data Kit (ODK) electronic data collection tool. The questionnaires were uploaded to smartphones and were pretested in the field before deployment. Clinicians administered household and individual human questionnaires, while veterinarians administered the herd-level and individual animal questionnaire. The first questionnaire was administered to household heads to collect herd-level animal husbandry practices linked with the transmission of *Brucella* spp. such as herd size, breeding between herds, sharing of grazing areas and sale or introduction of new animals into the herd. Individual-level animal data of the recruited animals were then collected, including age, sex, species (cattle, camel, goat, or sheep) and history of reproductive syndromes (such as abortion, weak birth, stillbirth, infertility, and swollen testis).

Livestock species were categorized into different age groups based on their age range. The first category included suckling animals such as sheep and goats (0–3 months old) and cattle and camels (0–6 months old),- and these animals usually stay close to households. The second category included weaners or young adults—no longer dependent on milk and often separated from their mothers. This category had goats and sheep that were 7–12 months old and cattle and camels that were seven months to two years old (for cattle) and up to three years old (for camels). Animals that were above these age ranges were categorized as adult animals that are of reproductive age.

Finally, human subject-level data such as demographic information, history of febrile illness, consumption of raw milk and milk products, handling of animals and animal products and occupational practices (which may include herding, milking, slaughtering, and assisting in animal births) were collected.

## Laboratory testing

Livestock serum samples were tested using the ID Screen Brucellosis Serum Indirect Multi-species kit (Grabels, France) for the detection of anti-*Brucella* spp. IgG antibodies in bovine, ovine, caprine, and porcine samples [34]. Briefly, the 10 μl of serum samples were diluted using 190 μl of dilution buffer two and loaded onto the pre-coated ELISA reaction plate. The test plates were incubated at room temperature for 45 minutes, after which they were washed three times. This was followed by the addition of 100 μl of the conjugate solution before the plate was incubated at room temperature for 30 minutes. After three washes, 100 μl of the

substrate solution was added, and the plate was incubated in darkness for 15 minutes. The reaction was stopped using 100 µl of the stop solution, and immediately, the optical densities (ODs) of each well were determined at 450 nm wavelength. All the samples were run in duplicates with each plate comprising positive and negative control. The results for each plate were considered valid if the mean OD for the positive control was greater than 0.35 and the ratio of the OD values between the positive and negative controls was greater than 3. The percentage S/P was calculated for each sample as described by the manufacturer. Following the manufacturer's recommended cut-off values, all samples whose percentage S/P was greater than or equal to 120% were considered positive, negative if less than or equal to 110% and doubtful if greater than 110% but less than 120%.

All human serum samples were tested using IBL-America IgG enzyme-linked immunosorbent assay (ELISA) kits (IBL-America, Minneapolis, USA) according to the manufacturer's recommendation [35]. Briefly, the sera were diluted at 1:100 using sample diluent and added onto antigen pre-coated microtitre plates followed by incubation for 1 hour at 37°C. Thereafter, 100 µl of ready to use conjugate was added and incubated for 30 minutes at 37°C before adding para-Nitrophenyl Phosphate (pNPP) substrate and further incubated at 37°C for 30 minutes. The conjugate–substrate reaction was conducted by adding 100 µl of ready-to-use stop solution included in the kit. The test sera ODs were read at 405 nm against a substrate blank and 630 nm was used as a reference wavelength to correct for the background fluorescence. Wash steps preceded both conjugate and substrate addition steps, while all sera and controls were run in duplicates. The results for each plate were valid when the substrate blank OD was less than 0.25 and the negative control gave negative results as per the interpretations. We first calculated the upper and lower cut-off limits to interpret our results. The upper cut-off limit was calculated as a product of the serum OD, the kit-specific correction factor (0.666) and the kit-specific grey zone given as 14% of the mean standard OD divided by Positive control OD. To calculate the lower cut-off limit, the kit specific grey zone was subtracted from the product of sample OD and the kit specific correction factor. Samples were interpreted as positive if their mean OD fell above the upper cut-off limit, doubtful if between the upper and lower cut-off limits, and negative if below the lower cut-off limit. Doubtful samples were considered as negative in subsequent analyses.

## Data management and statistical analyses

The serological and questionnaire data were first combined into one comma-delimited value (CSV) file and cleaned prior to analysis. We performed all analyses using the R statistical environment, version 3.6.0 [36], with descriptive statistics being analyzed using the *gmodels* [37] and *epiR* [37] packages. The first analysis was the calculation of animal/subject and household/herd levels overall seroprevalence estimates of *Brucella* spp., with a 95% confidence interval (CI) in humans and livestock. Seroprevalence was assessed at the individual, household, and herd levels. A household or herd was considered seropositive if at least one human or animal, respectively, tested positive. Additional analysis of livestock data included the estimation of animal/subject seroprevalence by categorical variables, which included animal species (camel, cattle, goat, and sheep), animal sex (male and female), animal age (suckling/lamb/calf, weaner/youngling, and adult) and herd size (<200, 200–400,>400). For human data, seroprevalence was estimated for gender (male and female), age (4–20,21–40,41–60,>60), and the highest level of education completed (no education, primary, secondary, and above). Herd size and age for humans were considered as quantitative discontinuous variables. To determine the normality of our sample data, we initially performed the Shapiro-Wilk test to check for the normality of residuals. These two variables violated the linearity assumption and were transformed into both

log-transformed and categorical variables and then compared with the outputs. We also determined the correlation between herd level exposure to *Brucella* spp. with human exposure. We defined an exposed herd as any herd that had at least one seropositive animal.

To start the statistical analyses, we employed Chi-square (χ2) test to assess the independence of categorical variables. Additionally, we conducted univariable mixed-effects logistic regression models to investigate the association between independent variables and the outcome variable. We used the generalised linear mixed models to conduct both univariable and multivariable analyses. All variables with a p-value lower than 0.20 from the univariable analysis were selected for the multivariable analysis [38]. The *lme4* package was used to fit the human and livestock data using the *glmer* function with random effects for household ID [38]. The final model was done through backward elimination. Firstly, unconditional models without independent variables were fitted for the livestock and human data, and then global models for variables selected from the univariable data. We systematically eliminated variables with p values lower than 0.05 until we achieved the lowest Akaike's information criteria (AIC) in the multivariable model. The final models from this procedure were then assessed for the goodness of fit through inspection of residual plots against the fitted values from the model [39]. Finally, we tested for potential interaction effects within the selected variables by conducting a likelihood ratio test (LRT) that was used to determine if there were potential interaction effects among product pairs of the factors in the final multivariable model [40]. Furthermore, we estimated the ICC for within-herd and within-household clustering of animal and human brucellosis using the *icc* function available in the performance package [41]

## Results

### Descriptive statistics

A total of 2,157 animals (258 cattle, 953 goats, 849 sheep and 97 camels) were sampled from 231 households and tested for anti-*Brucella* antibodies against *Brucella* spp. In eleven households (3.0%), animals were not available for sampling. In Garbatula, Sericho and Kinna wards, 977 (45.1%), 551 (25.5%) and 629 (29.2%) animals were sampled, respectively. Of the total sample, 1660 (77.0%) were female and 497 (23.0%) were males. The median herd size, combining cattle, sheep, goats, and camels was 176 (range: 8–1113), while the median herd/flock sizes for cattle, goats, sheep, and camels were 80 (range: 1–460), 80 (range: 8–546), 79 (range: 4–626) and 50 (range: 19–100), respectively. The median number of total animals sampled per household was 10 (range: 1–16).

This study also sampled 683 humans from 242 households. Most humans were sampled in Garbatula ward (39.1%, n = 267), followed by Sericho ward (34.8%, n = 238) and Kinna ward (26.1%, n=178). More males (83.0%, n = 567) were sampled than females (17.0%, n = 116). The median age of sampled subjects was 34 (range: 4–82), while the median number of individuals sampled in each household was 3 (range: 1–8).

### Seroprevalence of *Brucella* spp., in livestock and humans and univariable mixed-effects logistic regression model results from the analysis of livestock and human data

The univariable mixed-effects logistic regression model results from the analysis of livestock data, with households as random effects, are shown in Table 1. The overall animal and herd-level seroprevalence estimates of *Brucella* spp. in livestock were 10.4% (95% CI: 9.2–11.7) and 51.7% (95% CI: 47.9–55.7), respectively. The animal-level seroprevalence differed among livestock species. Camels had the highest seroprevalence of 19.6% (95% CI: 12.4–27.3), followed

**Table 1. Univariable mixed-effects logistic regression analysis of the factors associated with livestock brucellosis seropositivity.**

| Variable | Category | N | Seroprevalence % (95% CI) | Odds Ratio (95% CI) | P-value |
|---|---|---|---|---|---|
| Animal seroprevalence | | 2157 | 10.4 (9.2–11.7) | | |
| Herd seroprevalence | | 231 | 51.7% (47.9–55.7) | | |
| Sex | Female | 1660 | 11.6 (10.2–13.2) | 1.0 (Ref.) | |
| | Male | 497 | 6.2 (4.4–8.4) | 0.5 (0.3–0.7) | 0.001 |
| Animal Species | Goats | 953 | 13.2 (9.3–17.1) | 1.0 (Ref) | |
| | Sheep | 849 | 5.4 (4.0–6.9) | 0.4 (0.2–0.5) | <0.001 |
| | Cattle | 258 | 13.1 (11.1–15.3) | 0.9 (0.6–1.5) | 0.020 |
| | camel | 97 | 19.6 (12.4–27.3) | 1.8 (0.9–3.6) | 0.020 |
| Animal age | Calves/suckling | 72 | 2. 8 (0.0–5.7) | 1.0 (Ref.) | |
| | Young adults | 292 | 3.1 (1.4–4.8) | 1.2 (0.2–5.8) | 0.853 |
| | Adults | 1793 | 11.9 (10.4–13.4) | 5.0 (1.2–21.5) | 0.030 |
| History of retained placenta | No | 1368 | 12.8 (11.1–14.6) | 1.0 (Ref.) | |
| | Yes | 87 | 10.3 (5.8–17.2) | 0.9 (0.4–2.2) | 0.986 |
| History of abortion | No | 1107 | 11.2 (9.5–13.1) | 1.0 (Ref.) | |
| | Yes | 348 | 17.2 (13.5–21.2) | 1.9 (1.3–2.8) | 0.001 |
| Introduction of animals into the herd | No | 1614 | 11.5 (10.0–13.0) | 1.0 (Ref.) | |
| | Yes | 543 | 7.2 (5.2–9.2) | 0.6 (0.4–0.9) | 0.024 |
| Sharing grazing area with wildlife | No | 302 | 11.3 (8.0–14.7) | 1.0 (Ref.) | |
| | Yes | 1855 | 10.2 (9.0–11.6) | 0.8 (0.5–1.4) | 0.534 |
| Breeding between herds | No | 960 | 11.6 (9.7–13.6) | 1.0 (Ref.) | |
| | Yes | 1197 | 9.4 (7.9–11.0) | 0.8 (0.6–1.2) | 0.306 |
| Herd size | 1–200 | 1247 | 9.1 (7.6–10.7) | 1.0 (Ref.) | |
| | 201–400 | 643 | 10.0 (7.8–12.2) | 1.1 (0.7–1.7) | 0.692 |
| | >400 | 267 | 17.2 (13.1–21.9) | 2.0 (1.1–3.4) | 0.014 |
| Sampling location | Kinna | 629 | 12.7 (10.3–15.4) | 1.0 (Ref.) | |
| | Garbatula | 977 | 7.5 (5.9–9.1) | 0.5 (0.3–0.8) | 0.006 |
| | Sericho | 551 | 12.9 (-10.3–15.7) | 1.0 (0.6–1.6) | 0.922 |

CI confidence interval; Ref., reference category.

N: number of animals sampled in each category

by goats at 13.2% (95% CI: 9.3–17.1), cattle at 13.1% (95% CI: 11.1–15.3) and sheep at 5.4% (95% CI: 4.0–6.9) in decreasing order (Table 1). Female animals had higher seroprevalence of 11.6% (95% CI: 10.2–13.2) compared to male animals with a prevalence of 6.2% (95% CI: 4.4–8.4). When stratified by age, the seroprevalence increased with age with adult animals having a higher seroprevalence of 11.9% (95% CI: 10.4–13.4) compared to young adults at 3.1% (95% CI: 1.4–4.8) and calves 2.8% (95% CI: 0–5.7). Animals from Sericho ward had a higher seroprevalence estimate of 12.9% (95% CI: 10.3–15.7) compared to Kinna at 12.7% (95% CI: 10.3–15.4) and least in Garbatula at 7.5% (95% CI: 5.9–9.1). Herds with more than 400 animals had higher odds of seropositivity compared to animals from smaller herds. Additionally, herds, where new animals were introduced, reported higher odds of being seropositive compared to herds that did not introduce new animals. Finally, herds with at least two animal species had significantly higher odds of exposure to *Brucella* spp. compared to those with one species. The results are given in Table 1.

Results for the human univariable models are shown in Table 2. The individual-level seroprevalence of *Brucella* spp. in humans was 54.0% (95% CI: 50.2–58.0) while the household seroprevalence was 86.4% (95% CI: 84.0–89.0). Exposure to *Brucella* spp. was significantly

**Table 2. Univariable mixed-effects logistic regression analysis of the factors associated with human brucellosis seropositivity.**

| Variable | Category | N | Seroprevalence % (95% CI) | Odds Ratio (95% CI) | P-value |
|---|---|---|---|---|---|
| Human seroprevalence | | 683 | 54.0% (95 CI: 50.2–58.0) | | |
| Household Seroprevalence | | 242 | 86.4% (95% CI: 84.0–89.0) | | |
| Gender | female | 116 | 40.5 (31.9–49.9) | 1 .0 (Ref.) | |
| | male | 567 | 56.8 (52.7–61.2) | 2.0 (1.3–3.1) | 0.002 |
| Age groups | ≤20 | 155 | 47.7 (40–56.0) | 1.0 (Ref.) | |
| | 21–40 | 269 | 54.7 (48.7–60.9) | 1.4 (0.9–2.1) | 0.150 |
| | 41–60 | 177 | 60.5 (53.7–68.3) | 1.7 (1.1–2.7) | 0.021 |
| | >60 | 82 | 50.0 (40.2–61.9) | 1.2 (0.7–2.0) | 0.627 |
| Education level | Secondary and above | 46 | 31.6 (21.7–47.6) | 1.0 (Ref.) | |
| | Primary | 117 | 47.0 (38.5–56.8) | 1.9 (0.9–4.0) | 0.094 |
| | No education | 520 | 57.5 (53.3–62.1) | 2.9 (1.5–5.7) | 0.002 |
| Sampling locations | Kinna | 178 | 47.2 (39.9–54.9) | 1.0 (Ref.) | |
| | Garbatula | 267 | 52.4 (46.4–58.8) | 1.2 (0.8–1.9) | 0.302 |
| | Sericho | 238 | 60.9 (54.6–67.2) | 1.8 (1.2–2.8) | 0.009 |
| Reported febrile illness in last 6 months | No | 458 | 51.8 (47.2–56.6) | 1.0 (Ref.) | |
| | Yes | 225 | 58.7 (52.4–65.6) | 1.4 (1.0–1.9) | 0.075 |
| Consumed milk (pasteurised or raw) | No | 8 | 37.5 (12.5–70.9) | 1.0 (Ref.) | |
| | Yes | 675 | 54.2 (50.4–58.2) | 2.1 (0.5–9.7) | 0.336 |
| Consumed raw milk (any of the species) | No | 43 | 32.6 (20.9–47.8) | 1.0 (Ref.) | |
| | Yes | 640 | 55.5 (51.6–59.6) | 2.6 (1.3–5.2) | 0.007 |
| Handled animals | No | 23 | 13.0 (4.4–27.5) | 1.0 (Ref.) | |
| | Yes | 660 | 55.5 (51.7–59.5) | 8.7 (2.5–30.5) | 0.001 |
| Handled manure | No | 385 | 55.3 (50.4–60.6) | 1.0 (Ref.) | |
| | Yes | 298 | 52.4 (46.6–58.3) | 0.9 (0.6–1.2) | 0.464 |
| Handled raw meat | No | 165 | 48.5 (41.2–56.8) | 1.0 (Ref.) | |
| | Yes | 518 | 55.8 (51.5–60.4) | 1.4 (0.9–2.0) | 0.112 |
| Consumed bush meat. | No | 531 | 52.9 (48.6–57.4) | 1.0 (Ref.) | |
| | Yes | 152 | 57.9 (50–65.9) | 1.3 (0.8–1.9) | 0.252 |
| Herd exposure | Non-exposed | 330 | 48.2 (42.7–53.9) | 1.0(Ref) | |
| | Exposed | 353 | 59.5 (54.4–64.9) | 1.6 (1.2–2.3) | 0.005 |

CI confidence interval; Ref., reference category.

N: number of animals sampled in each category.

higher in males 56.8% (95% CI: 52.7–61.2) compared to females 40.5% (95% CI: 31.9–49.9). When data were stratified by age, the individual human seroprevalence estimate increased with age. On education level, higher seroprevalence was among those who had no formal education at 57.5% (95% CI: 53.3–62.1) followed by those who completed primary education at 47.0% (95% CI: 38.5–56.8) and it was least among those attained secondary or higher education at 31.6% (95% CI: 21.7–47.6). Residents of Sericho had a higher seroprevalence of 60.9% (95% CI: 54.6–67.2) compared to those in Garbatula at 52.4% (95% CI: 46.4–58.8) and Kinna at 47.2% (95% CI: 39.9–54.9). Among the risk factors for exposure to *Brucella* spp. analyzed, consuming raw milk from any species had higher odds of seropositivity than consuming pasteurized milk. Additionally, handling livestock was identified as a significant risk factor, but we found no significant association with *Brucella* spp. exposure among those who handled wild animals or consumed bushmeat.

## Multivariable analyses

The results from the multivariable mixed-effects model for animal data are summarized in Table 3. Four fixed effects were significant predictors of *Brucella* spp. exposure among animals, including animal sex, age, species, and herd size. Specifically, female animals had 1.7 times higher odds of being seropositive compared to male animals. The odds of being seropositive increased with increasing animal age, and adult animals were seven times more likely to be exposed to *Brucell*a spp. than calves. Sheep had significantly lower odds of being seropositive compared to goats. However, cattle or camel did not have a significant effect on the likelihood of exposure when compared to goats. Animals in larger herd sizes (above 400 animals) had significantly higher odds of *Brucella* spp. exposure compared to animals in smaller herds, with the likelihood of exposure increasing with the increase in the size of the herds. The history of abortion variable in sampled animals was excluded from the multivariable models because it only affected adult females and would not improve the model's predictability, despite its significant association with exposure to brucellosis in the univariable analysis. The ICC estimate for the household random effect in the final model was 0.138 (95% CI: 0.554–1.046), while the LRT χ2 for the product pairs created for the independent variables in the final models was not significant (p ≤0.05), suggesting that the model's predictors were not significantly collinear.

The outputs from the multivariable mixed-effects model for human data are summarized in Table 4. Three variables were significant: gender, education level, and herd exposure. Males had twice the odds of exposure compared to females, while individuals who did not complete any formal education had three times higher odds of exposure compared to those who completed secondary education or higher. The odds of exposure decreased with an increase in educational attainment.

Consumption of raw milk did not significantly affect the likelihood of exposure. There was a strong correlation between human seropositivity and livestock seropositivity for *Brucella* spp., with almost two-fold higher odds of exposure to people from households with at least one

**Table 3. Multivariable mixed-effects logistic regression analysis of the factors associated with livestock brucellosis seropositivity.**

| Variable | Category | Odds Ratio (95% CI) | SE | Z | P- value |
|---|---|---|---|---|---|
| Fixed effects | | | | | |
| Sex | Male | 1.0 (Ref.) | | | |
| | Female | 1.6 (1.1–2.5) | 0.217 | 2.195 | 0.028 |
| Age category | Calves | 1.0 (Ref.) | | | |
| | Weaners | 1.6 (0.2–8.2) | 0.827 | 0.591 | 0.554 |
| | Adults | 7.6 (1.7–33.8) | 0.763 | 2.655 | 0.008 |
| Animal species | Goats | 1.0 (Ref.) | | | |
| | Cattle | 1.3 (0.8–2.1) | 0.254 | 0.992 | 0.321 |
| | Camel | 2.4 (1.2–4.8) | 0.355 | 2.470 | 0.140 |
| | Sheep | 0.3 (0.2–0.5) | 0.189 | -5.587 | <0.001 |
| Herd size | 1–200 | 1.0 (Ref.) | | | |
| | 201–400 | 1.2 (0.7–1.8) | 0.220 | 0.644 | 0.520 |
| | Above 400 | 2.0 (1.2–3.5) | 0.275 | 2.611 | 0.009 |

Ref., reference category; CI, confidence interval; SE, standard error.

LRT-Likelihood-Ratio Test

Log likelihood = -655.55, number of observations = 2157, number of households = 231

The variance for Household ID as the random effect variable was 0.625 (95% CI: 0.5537–1.046), std dev = 0.79

**Table 4. Multivariable mixed-effects logistic regression analysis of the factors associated with human brucellosis seropositivity.**

| Variable | Category | Odds Ratio (95% CI | SE | Z | P-value |
|---|---|---|---|---|---|
| Fixed effects | | | | | |
| Gender | Female | 1 (Ref) | | | |
| | Male | 2.0 (1.3–3.1) | 0.224 | 3.042 | 0.002 |
| Handling animals | No | 1 (Ref) | | | |
| | Yes | 6.2 (1.7–22.3) | 0.172 | 2.813 | 0.005 |
| Highest Education level completed | Secondary and above | 1 (Ref) | | | |
| | Primary | 1.8 (0.9–3.9) | 0.385 | 1.574 | 0.103 |
| | No education | 2.9 (1.5–5.8) | 0.347 | 3.092 | 0.002 |
| Herd exposure | No | 1 (Ref) | | | |
| | Yes | 1.4 (1.0–2.0) | 0.171 | 2.127 | 0.033 |
| Location | Kinna | 1 (Ref) | | | |
| | Garbatulla | 1.3(0.9–2.0) | 0.213 | 1.356 | 0.175 |
| | Sericho | 1.6(1.1–2.5) | 0.220 | 2.252 | 0.024 |

Ref., reference category; CI, confidence interval; SE, standard error.

Log likelihood = -445.5, number of observations = 683, number of households = 242

The variance for Household ID as the random effect variable was 0.126 (95% CI: 0.00–0.744), SE 0.764

seropositive animal than those without. The ICC value calculated from the variance estimates of the final multivariable model was 0.03 (95% CI; 0.01–0.74).

## Discussion

This linked study aimed to estimate the seroprevalence of *Brucella* spp. among humans and animals from the same household and to determine the associated risk factors with *Brucella* spp. seropositivity in a pastoralist area in northern Kenya.

The study found a high level of brucellosis seroprevalence at the individual human and household levels, with individual seroprevalence of 54% (95% CI: 50.2–58.0) and household seroprevalence of 86.4% (95% CI: 84.0–89.0). The individual seroprevalence of 54% is higher than the 14–46.5% range reported in comparable studies conducted among pastoralist communities [23, 24, 26, 42]. These high seroprevalence estimates could be due to the study participants' constant exposure to infected animals and animal products. Additionally, the study sampled more men and older individuals (mean age of 34), compared to other studies [23, 24], and men are more prone to infection compared to females due to occupational exposure, while older individuals have had a longer exposure time to the disease [3].

We observed high animal- and herd-level seroprevalence estimates of *Brucella* spp. in livestock. The observed overall seroprevalence was lower than that reported in seroprevalence studies in Marsabit (19.2%) and Baringo (22.3%) counties in Kenya [23, 25, 26] but higher than reported in Garissa County (3.5%) in Kenya [24] and the Somali region of Ethiopia (0.3%) [42]. The seroprevalence varied by species, with camels having the highest seroprevalence, followed closely by goats and cattle, while sheep exhibited the lowest seroprevalence. The high seroprevalence in camels has also been observed in studies conducted in Isiolo County [43, 44] , while studies in neighbouring Marsabit County have reported lower estimates [23, 26]. Although camels are not the primary hosts for *Brucella* spp., the high seroprevalence detected in this study suggests their potential role in maintaining brucellosis transmission in the area. The Borana community in Isiolo County traditionally keeps cattle, goats, and sheep but is increasingly embracing camels due to their resilience to climate

variability [45, 46]. The lower seroprevalence estimates in sheep have been reported in other studies [24, 43], which utilised similar testing strategies, but there is currently limited evidence on why sheep have lower *Brucella* exposure levels compared to other ruminant species such as goats and cattle.

The study utilized multivariable models to identify factors associated with *Brucella* spp. seropositivity in livestock. Female animals had almost two-fold higher odds of exposure than males. One main reason for this high likelihood is increased brucellosis transmission through reproductive fluids. Female animals can be exposed to bacteria during breeding, pregnancy, and birth, making them more vulnerable to infection [47]. Additionally, female animals are kept longer for reproduction and milk production, which is a common practice in pastoralist communities [23, 24, 26]. In contrast, male animals are typically sold to meet the household's needs, such as food, healthcare, veterinary services, and education.

Adult animals were found to be more than seven times as likely as calves to be exposed to brucellosis. Older animals have greater exposure time, travel further and therefore have more opportunities to mix with other herds, leading to increased disease transmission. Similar findings have been reported in Marsabit County and Garissa County in Kenya [23, 24]. Further, the study found that female animals that had experienced abortion in the previous year were more likely to be seropositive for brucellosis at the univariable analysis. However, this factor was not included in the multivariable analysis, as it affected only adult female animals. Nonetheless, this finding emphasizes the well-known association between spontaneous abortion in animals and brucellosis infection [48, 49]. Despite the established link between brucellosis and spontaneous abortion in livestock, pastoralists may not always associate spontaneous abortions in their animals with brucellosis [50]. As a result, they may continuously expose themselves to the disease while handling infected abortion materials.

Brucellosis prevalence was highly correlated with herd size, with larger herds having a significantly higher risk of exposure to the disease. Large herds are often associated with poor sanitation, clustering of animals, and mixing of animals from different herds and species. Several studies in Africa have found a similar correlation between brucellosis seropositivity and herd size [24, 49, 51].

The multivariable model on humans found several risk factors to be associated with brucellosis at the individual level. Males were found to have a higher seroprevalence than women, with a seropositivity rate of 57% compared to 41% in females. Men had two-fold higher odds of exposure compared to women. Worldwide, brucellosis is more frequently detected in males than females. This difference in odds of infection between the genders can be attributed to occupational exposure, such as frequent contact with infected animals or contaminated animal products while herding, milking, slaughtering, or assisting with animal birth [3]. Similar findings have been documented in local studies [23, 24]. Additionally, we found that individuals who regularly handled animals (participated in herding, milking, slaughtering, or assisting in animal births) had six times higher odds of being seropositive than those who did not. These findings highlight the importance of proper hygiene practices when working with animals or animal products and the need for education and awareness campaigns to reduce the risk of brucellosis transmission in high-risk populations, such as male herders.

The consumption of milk and milk products is a crucial aspect of the diet of Borana pastoralists, but it can also pose serious health risks, particularly when consumed raw. Despite the risks, a vast majority (94%) of our study participants reported consuming raw milk in the three months preceding the study. However, regular consumption of raw milk was not a significant risk factor in the multivariable analysis. We believe this is because almost all participants consume raw milk. On the contrary, other studies among pastoralist communities in Isiolo and Marsabit County have linked the consumption of raw milk to brucellosis exposure

[23, 52]. Boiling milk is a simple and effective way to kill milk-borne pathogens including *Brucella* spp. while preserving the nutritional value of the milk. However, a previous study in southern Ethiopia found that Borana pastoralists considered boiling milk to reduce nutrient value and make the milk lack taste [53], emphasizing the need to find a balance between preserving cultural practices and minimizing health risks.

The poor literacy levels in the study area were reflected in the fact that most participants had not received any formal education. Attaining secondary education or higher was found to be protective against exposure to *Brucella* spp. and those with no formal education had about three times higher odds of exposure to brucellosis. This finding aligns with previous observations by a study conducted in neighboring Marsabit county [23] , which is also inhabited by the Borana community. These results have important implications for public health interventions aimed at creating awareness about brucellosis. Higher education levels are associated with better awareness of the disease [54, 55], and therefore, designing culturally sensitive, simple, and accessible public education measures would be critical to reducing disease transmission among less educated pastoralists.

Human exposure to brucellosis had a 1.5 times higher likelihood of occurring within households that had a positive herd. This shows that many human infections are transmitted through close contact with the household's infected animal materials. This finding is consistent with other studies conducted in neighbouring areas [23, 24]. However, the correlation between human and animal *Brucella* spp. positivity was much lower in this study, suggesting that the risk of transmission at the household level was lower in our study area. This is supported by the low intra-class correlation (ICC) of 0.03 in humans, which indicates that human exposure to *Brucella* spp. was not correlated among individuals within households. This is not surprising because human-to-human transmission of brucellosis is rare, and most infections are typically transmitted through the consumption of contaminated animal products or contact with infected animals.

Finally, our data reveal that individuals living in Sericho ward have significantly higher *Brucella* spp. seropositivity rates compared to those in Kinna and Garbatula Wards. Sericho is a remote, low-lying area preferred by nomadic pastoralists for its flat terrain and extensive grazing lands. However, unlike Kinna and Garbatula, where many pastoralists live in settlements with some amenities such as schools and health facilities, Sericho is quite underserved, and most people have limited access to healthcare and education [30], which may explain the higher seropositivity rates. Additional research is necessary to understand better the reasons behind the higher seropositivity rates in Sericho ward.

## Limitations

Our study had a few limitations. The study was conducted during a hot and dry season, a period often characterized by adaptation strategies, including an increase in droughts-adapting livestock, herd splitting, household splitting and clustering of herds for security reasons and in areas proximal to pasture and water for their livestock. [56, 57]. Additionally, the study included a few animals below six months of age, which have the potential to carry maternal antibodies. This introduced a small risk of obtaining false positive results. Finally, the study employed ELISA exclusively for serological testing, despite its sensitivity and specificity being subject to fluctuations, particularly in endemic areas. Moreover, ELISA is susceptible to cross-reactivity with closely related pathogens, potentially resulting in false-positive outcomes. Additionally, the test may fail to detect active infections, thereby resulting in false-negative results [35, 58, 59]. To enhance accuracy, it is advisable to combine ELISA with other serological tests,

such as the Rose Bengal test (RBT) and the serum (tube) agglutination test (SAT), along with molecular techniques [6]

## Conclusion and recommendations

Our study reports a high prevalence of brucellosis in humans and livestock in the study area, underscoring the need for targeted prevention and control measures to curb its spread. The study revealed several factors associated with *Brucella* spp. seropositivity in humans and livestock, including animal age, herd size, livestock species, gender, education level, and herd exposure. Enhancing animal health management procedures, encouraging appropriate hygiene standards when handling animals, and promoting the consumption of safe animal products are three vital ways of mitigating disease transmission. This can involve creating more public awareness about the disease to minimize the risk of disease transmission and increasing veterinary outreach services by the County Government of Isiolo to pastoralists, such as vaccination. In addition, the One Health approach applied in this study emphasizes the significance of understanding the interactions between humans and animals in transmitting and controlling zoonotic diseases, showing the need to establish an integrated One Health surveillance system in such high-risk populations. Future research should examine the efficacy of measures to mitigate the spread of brucellosis and the socio-economic impacts of brucellosis in the region.

## Supporting information

**S1 Data. Animal dataset for analysis of seroprevalence related risk factors of Brucella spp. in livestock and humans in Garbatula subcounty, Isiolo county, Kenya.**
(XLSX)

**S2 Data. Human dataset for analysis of seroprevalence related risk factors of Brucella spp. in livestock and humans in Garbatula subcounty, Isiolo county, Kenya.**
(XLSX)

## Acknowledgments

We thank all people of Garbatula subcounty for accepting to join the study. We also thank the public and animal health officials from Isiolo County for approving the study to be implemented in their area. Additionally, we thank the human and animal health personnel who assisted in sampling study participants.

The content of the information does not necessarily reflect the position or the policy of the federal government, and no official endorsement should be inferred.

## Author Contributions

**Conceptualization:** Athman Mwatondo, Mathew Muturi, James Akoko, Stephen Gichuhi, Marianne W. Mureithi, Bernard Bett.

**Data curation:** Athman Mwatondo, Mathew Muturi, James Akoko, Daniel Nthiwa, Bernard Bett.

**Formal analysis:** Athman Mwatondo, Mathew Muturi, James Akoko, Daniel Nthiwa, Bernard Bett.

**Funding acquisition:** Bernard Bett.

**Investigation:** Athman Mwatondo, Mathew Muturi, James Akoko, Richard Nyamota, Josphat Maina, Jack Omolo, Stephen Gichuhi, Marianne W. Mureithi, Bernard Bett.

**Methodology:** Athman Mwatondo, Mathew Muturi, Richard Nyamota, Daniel Nthiwa, Josphat Maina, Jack Omolo, Stephen Gichuhi, Marianne W. Mureithi, Bernard Bett.

**Project administration:** Athman Mwatondo, James Akoko.

**Resources:** Athman Mwatondo, Bernard Bett.

**Software:** Athman Mwatondo, James Akoko, Bernard Bett.

**Supervision:** Stephen Gichuhi, Marianne W. Mureithi, Bernard Bett.

**Validation:** Athman Mwatondo, Mathew Muturi, James Akoko, Richard Nyamota, Marianne W. Mureithi, Bernard Bett.

**Visualization:** Athman Mwatondo, Mathew Muturi, James Akoko, Stephen Gichuhi, Bernard Bett.

**Writing – original draft:** Athman Mwatondo.

**Writing – review & editing:** Athman Mwatondo, Mathew Muturi, James Akoko, Richard Nyamota, Daniel Nthiwa, Josphat Maina, Jack Omolo, Stephen Gichuhi, Marianne W. Mureithi, Bernard Bett.

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
