## [Decision Letter · Decision Letter 0]

19 Jun 2023

Dear Dr. Mwatondo,

Thank you very much for submitting your manuscript "Seroprevalence of Brucella spp. in livestock and humans in Isiolo county in Kenya" for consideration at PLOS Neglected Tropical Diseases. As with all papers reviewed by the journal, your manuscript was reviewed by members of the editorial board and by several independent reviewers. In light of the reviews (below this email), we would like to invite the resubmission of a significantly-revised version that takes into account the reviewers' comments. 

Please respond to the attached reviewer comments

We cannot make any decision about publication until we have seen the revised manuscript and your response to the reviewers' comments. Your revised manuscript is also likely to be sent to reviewers for further evaluation.

Sincerely,

Georgios Pappas

Academic Editor

Ana LTO Nascimento

Section Editor

Please respond to the attached reviewer comments

Reviewer's Responses to Questions

**Key Review Criteria Required for Acceptance?**

**Methods**

-Are the objectives of the study clearly articulated with a clear testable hypothesis stated?

-Is the study design appropriate to address the stated objectives?

-Is the population clearly described and appropriate for the hypothesis being tested?

-Is the sample size sufficient to ensure adequate power to address the hypothesis being tested?

-Were correct statistical analysis used to support conclusions?

-Are there concerns about ethical or regulatory requirements being met?

Reviewer #1: yes

yes

yes

yes

yes

I have not seen ethical declaration

Reviewer #2: The authors articulated the objectives of the hypothesis tested. The authors have appropriate study design and clearly defined the population studied. Ethical requirements were met.

Reviewer #3: (No Response)

**Results**

-Does the analysis presented match the analysis plan?

-Are the results clearly and completely presented?

-Are the figures (Tables, Images) of sufficient quality for clarity?

Reviewer #1: Yes

moderate, need modification

yes but some tables are long

Reviewer #2: The results were appropriately presented. The figures and tables were presented accordingly.

Reviewer #3: (No Response)

**Conclusions**

-Are the conclusions supported by the data presented?

-Are the limitations of analysis clearly described?

-Do the authors discuss how these data can be helpful to advance our understanding of the topic under study?

-Is public health relevance addressed?

Reviewer #1: yes

yes

yes

yes

Reviewer #2: The conclusions were supported by the data presented. The limitations were described clearly. The usefulness of the data was presented

Reviewer #3: (No Response)

**Editorial and Data Presentation Modifications?**

Reviewer #1: The manuscript can be published after correction of the comments provided.

Reviewer #2: The manuscript can be accepted after the suggested minor corrections.

Reviewer #3: (No Response)

**Summary and General Comments**

Reviewer #1: The study provided important information for the readers. Minor modification required at presentation of the results

Reviewer #2: Can the authors give reasons why Rose Bengal test (RBT) was not used for screening?, or for adaptation of brucellosis testing in series which was not performed.

How would the authors explain the elimination of false positives because of the widely used serologic ELISAs for the detection of brucellosis which is very high, to the extent that some laboratories apply not only the CFT, but parallel with it, the classic Rose Bengal (RBT) or/and serum agglutination (SAT) tests for confirmation. 

There is the possibility of cross-reaction of Gram-negative bacterium species or other unknown organisms or molecular antigens that are generally considered as contributing in false positives using ELISA, how would the authors clarify this in this study.

Serology alone can only be a presumptive test since other pathogens (for example, Y.

337 enterocolitica, O:9) can cross-react with the tests, thereby leading to false-positive results. 

Can the authors elaborate more on the vaccination status or strategies in the study areas, as seropositivity could have been due to reaction to vaccination.

Reviewer #3: My general observation is that the manuscript is well written and the subject is important, so it should be considered after minor corrections as indicated in the track changes (attached)

PLOS authors have the option to publish the peer review history of their article (what does this mean?). If published, this will include your full peer review and any attached files.

Reviewer #1: No

Reviewer #2: No

Reviewer #3: Yes: Coletha Mathew

Figure Files:

Data Requirements:

Please note that, as a condition of publication, PLOS' data policy requires that you make available all data used to draw the conclusions outlined in your manuscript. Data must be deposited in an appropriate repository, included within the body of the manuscript, or uploaded as supporting information. This includes all numerical values that were used to generate graphs, histograms etc.. For an example see here: http://www.plosbiology.org/article/info:doi%2F10.1371%2Fjournal.pbio.1001908#s5.
---

## [Decision Letter · Decision Letter 1]

26 Sep 2023

Dear Dr. Mwatondo, 

We are pleased to inform you that your manuscript 'Seroprevalence and related risk factors of Brucella spp. in livestock and humans in Garbatula subcounty, Isiolo county, Kenya' has been provisionally accepted for publication in PLOS Neglected Tropical Diseases.

Best regards,

Georgios Pappas

Academic Editor

Ana LTO Nascimento

Section Editor

-

Reviewer's Responses to Questions

**Key Review Criteria Required for Acceptance?**

**Methods**

-Are the objectives of the study clearly articulated with a clear testable hypothesis stated?

-Is the study design appropriate to address the stated objectives?

-Is the population clearly described and appropriate for the hypothesis being tested?

-Is the sample size sufficient to ensure adequate power to address the hypothesis being tested?

-Were correct statistical analysis used to support conclusions?

-Are there concerns about ethical or regulatory requirements being met?

Reviewer #2: the objectives are clearly articulated, the study design and study population are clearly described. the analysis and conclusions are sufficient

Reviewer #3: (No Response)

**Results**

-Does the analysis presented match the analysis plan?

-Are the results clearly and completely presented?

-Are the figures (Tables, Images) of sufficient quality for clarity?

Reviewer #2: the analysis match the plan, and results were clearly presented and with appropriate fgures and tables

Reviewer #3: (No Response)

**Conclusions**

-Are the conclusions supported by the data presented?

-Are the limitations of analysis clearly described?

-Do the authors discuss how these data can be helpful to advance our understanding of the topic under study?

-Is public health relevance addressed?

Reviewer #2: conclusions are supported by data and limitations were clearly presented. publich health significance was addressed.

Reviewer #3: (No Response)

**Editorial and Data Presentation Modifications?**

Reviewer #2: Accept the manuscript

Reviewer #3: (No Response)

**Summary and General Comments**

Reviewer #2: the manuscript is clearly articulated and presented and we benefit the scientific world.

Reviewer #3: The manuscript is now in order and of acceptable standard.

Just change the word "Brucella" in the title into Italics. This applies throughout the document

PLOS authors have the option to publish the peer review history of their article (what does this mean?). If published, this will include your full peer review and any attached files.

Reviewer #2: **Yes: **Francis Babaman Kolo

Reviewer #3: No

---

## [Editor Report · Acceptance letter]

9 Oct 2023

Dear Dr Mwatondo,

We are delighted to inform you that your manuscript, " Seroprevalence and related risk factors of Brucella spp. in livestock and humans in Garbatula subcounty, Isiolo county, Kenya ," has been formally accepted for publication in PLOS Neglected Tropical Diseases.

Best regards,

Shaden Kamhawi

co-Editor-in-Chief

Paul Brindley

co-Editor-in-Chief
